# Rheological Properties of Non-Adhesive Embolizing Compounds—The Key to Fine-Tuning Embolization Process-Modeling in Endovascular Surgery

**DOI:** 10.3390/polym15041060

**Published:** 2023-02-20

**Authors:** Iuliia Kuianova, Alexander Chupakhin, Alexey Besov, Anton Gorbatykh, Dmitry Kislitsin, Kirill Orlov, Daniil Parshin

**Affiliations:** 1Lavrentyev Institute of Hydrodynamics SB RAS, 630090 Novosibirsk, Russia; 2Meshalkin National Medical Research Center, 630055 Novosibirsk, Russia

**Keywords:** embolic agent, rheology, hemodynamics, arteriovenous malformation (AVM), CFD

## Abstract

The study of polymers’ rheological properties is of paramount importance both for the problems of their industrial production as well as for their practical application. Two polymers used for embolization of arteriovenous malformations (AVMs) are studied in this work: Onyx-18^®^ and Squid-12^®^. Viscosity curve tests and computational fluid dynamics (CFD) were used to uncover viscosity law as a function of shear rate as well as behavior of the polymers in catheter or pathological tissue models. The property of thermal activation of viscosity was demonstrated, namely, the law of dependence of viscosity on temperature in the range from 20 ${}^{{}^{\circ}}$C to 37 ${}^{{}^{\circ}}$C was established. A zone of viscosity nonmonotonicity was identified, and a physical interpretation of the dependence of the embolic polymers’ viscosity on the shear rate was given on the basis of Cisco’s model. The obtained empirical constants will be useful for researchers based on the CFD of AVMs. A description of the process of temperature activation of the embolic polymers’ viscosity is important for understanding the mechanics of the embolization process by practicing surgeons as well as for producing new prospective embolic agents.

## 1. Introduction

In modern vascular surgery, more and more preference is given to minimally-invasive interventions that are less traumatic for the patient [1]. One direction for such operations is that the embolization of cancer, arteriovenous malformations (AVMs), fistulas, etc. Depending on the type of pathology, its localization, stage and other medical aspects, embolization polymers differ, respectively, as adhesive and non-adhesive. Adhesive polymers are adhesives based on cyanoacrylate and are designed to cause occlusion, i.e., sealing of, relatively large vessels (fistulas). In contrast, non-adhesive composites are aimed at occluding relatively small vessels (racemose components of the anomaly). The main pathologies for which such polymers are used are tumors and arteriovenous malformations (AVMs). The issues involved in choosing the optimal technique and embolization protocol are actively discussed in the literature. First of all, this is the choice of parameters for the optimality of this process: determining the embolization access—arterial or venous [2], and determining the sequence of using certain composites during embolization in the presence of both the racemose and fistula components of the AVM [3]. The main problems in this case are determining the volume of polymer required for embolization [4], the optimal law of its administration, as well as the function of its consumption depending on time. In fact, these are the only parameters that the surgeon controls after choosing one or the other embolic polymer. This is a very important task of practical medicine; therefore, a large number of medical works are devoted to the discussion of these issues [5,6].

To solve such problems, in recent years, along with experimentation, mathematical modeling has been increasingly used [7,8,9], as well as some trials of promising embolisates [10].

Non-adhesive polymers and polymers based on microspheres are of greatest interest for the study of rheological properties [11]. A feature of non-adhesive drugs is that they do not stick to the vascular wall during precipitation; thus, they have an increased ability to spread in the vascular bed, better filling the blood vessels of pathological formations, and the possibility of a longer and more controlled administration. On the other hand, these substances have an increased tendency to migrate into those vessels of the body where they are undesirable and may be accompanied by clinical complications. This is especially evident in the treatment of high-flow arteriovenous shunts [12,13]. The penetration of such drugs into the venous bed can cause vein thrombosis [14,15] and deterioration of lung tissue perfusion, which can have a negative effect on the whole organism. For adhesive polymers, most of which are based on cyanoacrylate derivatives, the number of possible complications is significantly less and mainly relate to complications associated with catheterization [16]. However, the scope of such polymers is usually large arteries, tumors, and fistulas.

Despite the presence of a number of studies on the chemical and rheological characteristics of the considered embolisates [10,11,17], it is worth noting a rather weak connection in such works with the formulation of medical problems [18], as well as problems from the field of computational modeling [19]. In the works reviewed, little attention is paid to the different temperature ranges and different ranges of shear rates at which embolisates are operated. These seemingly purely medical problems are, in fact, of a fundamental nature and can be solved only with an integrated approach that combines the study of the rheological properties of embolic polymers, as non-Newtonian media, and the study of the laws of motion of such media along with a geometrically complex, branched network of channels of various sizes. Rheological properties of the embolic polymer can vary depending on network geometry. Moreover, it turns out that it is necessary to take into account the detailed thermodynamics of this process, which, unlike the classical problems of hydrodynamics, cannot be considered isothermal.

If, for solving problems of hemodynamics in large vessels, the generally accepted gold standard is the use of Newtonian fluid models, then for solving problems of embolization of a small branched network of vessels with a caliber of up to 1 mm, such an assumption is unacceptable. Non-Newtonian properties begin to manifest themselves significantly in the course of such polymers and play a decisive role. Some information about penetration depth score, the dependence of pressure in a catheter with respect to volume flow rate of the embolic agent, length of reflux and others is known [17,20]. In addition, the rheology of such polymers is complicated by the presence of a contrast media [21,22], which significantly affects the sedimentation of polymers in the vasculature with respect to the origin of the contrasting agent. Figure 1 shows varieties of iodine radiopaque contrast agents (top—nephrotropic, low osmolar iohexol [23]; bottom—iocarmic acid [24]). Since an iodine-containing contrast agent is capable of causing anaphylactic shock, tantalum-containing analogues have been developed. Figure 2 shows the synthesis process of carboxybetaine zwitterionic-tantalum oxide nanoparticles (CZ-TaO NPs) which includes three stages: saline condensation, end-group hydrolysis and purification. In this regard, a number of questions arise considering the rheology of the flow of embolization polymers depending on shear rate and temperature. The last factor is fundamental, since the syringe with the embolic polymer is located in the operating room outside the patient’s body, at a temperature of 20–25 ${}^{{}^{\circ}}$C degrees, and the filling of the anomaly node with the embolic polymer occurs inside the patient’s body and is carried out at a body temperature of 37 ${}^{{}^{\circ}}$C degrees.

In this paper, an experimental approach to the study of the rheological properties of embolization materials used in clinical practice was considered. To highlight shear rate values that correspond to the physiological state of a human body, a numerical model of AVM embolization with non-adhesive embolization polymers was verified based on rheological test data. In addition to their fundamental importance to the understanding flow mechanics, rheometry data can become the basis for constructing a phenomenological model of the viscosity of such polymers. Therefore, specific parameters are proposed for models of pseudoplastic and dilatant fluids that study the viscosity of embolic polymers both under laboratory (T = 20 ${}^{{}^{\circ}}$C) and surgical (T = 37 ${}^{{}^{\circ}}$C) conditions.

## 2. Materials and Methods

### 2.1. Materials

The embolic polymer material provided by the clinical partner of the research was studied. The material consists of irretrievable remnants of endovascular interventions. Approximately 0.5 mL of material was used for each of the tests. Since the manufacturer’s official instructions recommend shaking the embolic polymer vial for at least 15 min, this necessary procedure was performed before the test. Onyx^®^ and Squid^®^ are gelling solutions that are composed of PVA co-polymers in dimethyl sulphoxide (DMSO) with tantalum (see Figure 3). An ethylene vinyl alcohol co-polymer (48 mol/L of ethylene and 52 mol/L vinyl alcohol) was dissolved in DMSO (see Figure 4). Micronized tantalum powder was suspended in the liquid polymer/DMSO mixture to provide fluoroscopic visualization [25]. As can be seen from the experimental data (see Figure 5), the rheological properties of the considered polymers during their regular agitation and their refusal to agitate significantly change, not only quantitatively (which is typical for both polymers at shear rates exceeding 10 s${}^{-1}$), but also qualitatively, which is most typical for the Onyx-18^®^ polymer at shear rates less than 1 s${}^{-1}$. In the course of the experiments, the same polymer was used for the Onyx/Squid suspensions with mixed and not-mixed tantalum. In the first case, the bottle was agitated for 15 min; this way, a uniform distribution of the radiopaque contrast agent in the polymer was achieved. However, in the second case, agitation was not used. As a result, a transparent component of the embolic polymer, presented by a solution of vinyl alcohol in DMSO, remained in the test area. Due to this limitation, a number of long-term tests (such as frequency, requiring approximately 40 min to vary the spectrum of physiologically adequate particles) were not performed which would have been of interest for a detailed understanding of polymer rheology.

### 2.2. Experimental Protocol

The rheological properties of the embolization compositions were studied on an Anton Paar MCR302 rheometer (Austria) using the CP50-1 Cone-Plane measuring system (see Figure 6) with a diameter of 50 mm, minimum clearance of 0.1 mm and an angle of $1^{{}^{\circ}}$. A P-PTD200 Peltier heating system and an H-PTD200 Peltier active housing, designed to minimize temperature gradients in the samples and prevent their evaporation during the measurement process, were used. The specified accessories provide maintenance of the set temperature with an accuracy of $0.01{\;}^{{}^{\circ}}$C. Testing was carried out using viscosity curve tests of the rheometer software (RheoCompass 1.30.1164 Release). In the process of testing the studied samples, the dependence of effective viscosity of the embolization compositions on the shear rate at given temperatures (laboratory T = 20 ${}^{{}^{\circ}}$C and surgical T = 37 ${}^{{}^{\circ}}$C) was determined.

### 2.3. Numerical Simulations

Numerical modeling of the embolization polymers’ flow through the canal network (see Figure 7) was performed using the ANSYS 2020R2 software package (CFX, license LIH SB RAS). Since the time interval of the embolization effect (the process from the beginning of the introduction of a polymer-based drug to the achievement of the desired clinical effect) is from 20 to 40 min, it is reasonable to model it as a stationary course of the embolic polymer. At the first stage, this model includes a flow in a long, thin tube simulating a catheter through which the embolic polymer is delivered to the circulatory system; the flow in the network is first though diverging, branching vessels and, finally, in vessels that again converge into one vessel.

The catheter wall is considered to be a thin material, but having a non-zero thickness of 0.2 mm. The condition on the fluid wall has type–interface (to take into account heat flux) with no slip velocity condition. At the outlet from the fluid flow region, the condition of zero pressure was set.

The Navier–Stokes equation system for the flow of a viscous, incompressible fluid was used as the governing equation. Considering that $m_{\alpha }={\left(\rho u_{i}\Delta n_{i}\right)}_{\alpha }$ is the mass flow through the surface of the volume element, Navier–Stokes equations can be written in discrete form for solving the problem using the finite element method [28]:(1)$$\left\{\begin{array}{c} \sum_{\alpha }m_{\alpha }{\left(u_{j}\right)}_{\alpha }-\sum_{\alpha }{\left(\rho \mu \frac{\partial u_{i}}{\partial x_{j}}\Delta n_{i}\right)}_{\alpha }=-\sum_{\alpha }{\left(p\Delta n_{j}\right)}_{\alpha }\\ \sum_{\alpha }m_{\alpha }=0, \end{array}\right.$$
where ρ—density, μ—dynamic viscosity, *u*—flow rate and *n*—normal vector to the computational domain.

Temperature effects can be neglected for the case of a convergent–divergent channel, since these flow stages occur at body temperature. On the contrary, when liquid flows through a conductor, there is a transition between the flow.

To use the obtained dependence of the polymers’ viscosity on the shear rate in the polymer flow calculations in the AVM model configuration, it was necessary to approximate the experimental data. For this purpose, the viscosity graphs were divided into three different segments; on each segment, the function approximation was performed linearly Figure 8. The temperature of the polymer was assumed to be constant, equal to 37 ${}^{{}^{\circ}}$C. Thus, the temperature transition from laboratory conditions (20–25 ${}^{{}^{\circ}}$C) to physiological ones was not considered in this simulation option.

The next stage of modeling was the implementation of a numerical calculation of the embolic polymer flow in the catheter using settings similar to the operating room. It was assumed that the catheter consists of two parts—an outer one, which is located outside the patient’s body, and an inner one, which runs from the puncture site in the femoral artery to the AVM nidus. The simulation was performed for the second (internal) segment. A volume flow rate of about 0.6 mL/min was used for conditions at the inlet in this part of the catheter. A catheter with an inner diameter of 1 mm was considered, and the temperature of the supplied embolization polymer was 20 ${}^{{}^{\circ}}$C.

Since the polymer supply process can be considered quasi-stationary and the catheter diameter does not change, the shear rate can also be considered unchanged. Therefore, the viscosity of such a polymer must be considered as a function of temperature only at a shear rate of approximately 10 s${}^{-1}$ (Figure 9).

At the liquid boundary (contact zone with the stent), the condition for maintaining the heat flux was set [29]: (2)$$Q=-\underset{A}{\;\int \int \;}{\dot{q}}^{"}\;dA,$$
where
(3)$${\dot{q}}^{"}=-k\Delta T$$

*T*—heat flux, *k*—coefficient of thermal conductivity, and *A*—heat transfer area. Since more than 90% of the polymer solution is DMSO, the known thermal constants of this substance were used [30]: $k=0.2\;w\cdot m^{-1}\cdot K^{-1}$ and the specific heat coefficient c=0.47 cal/g/C.

## 3. Results

### 3.1. Experimental Results

For the purposes of this work, rheological tests were carried out in the cone-plane system (Figure 6). The tests were carried out for two models of embolic polymers that are currently used in clinical practice for AVM embolization.

Similar flow regimes (bleaching 20 ${}^{{}^{\circ}}$C and separately 37 ${}^{{}^{\circ}}$C) were implemented in two different regions—at the injection site of the embolization polymer into the catheter (corresponding to 20 ${}^{{}^{\circ}}$C) and in the AVM nidus (corresponding to 37 ${}^{{}^{\circ}}$C). However, in the area of the catheter penetration into the femoral artery, a temperature transition was considered. Therefore, it was useful to study the behavior of embolic polymers in the temperature range of 20–37 ${}^{{}^{\circ}}$C:

In a numerical analysis of the flow of viscous liquids at a temperature of 37 ${}^{{}^{\circ}}$C, which have a rheology similar to the tested samples of ONYX-18^®^ and SQUID-12^®^ (see Table 1, Figure 10), it turned out that the maximum pressure for ONYX-18^®^ -like liquid significantly exceeds the maximum pressure for SQUID-12^®^ -like liquid in the presence of a fistula, which is easily explained by the need of using more viscous polymers in the presence of large vessels in the AVM nidus.

At the same time, the maximum shear stresses for both polymers remain comparable (see Figure 11). From essential considerations, it may be assumed that the zone of embolic polymer applicability is determined by the local pressure peak, which is achieved for the Onyx flow at d = 1.5, and for the Squid flow at d = 1.75 (see Figure 12). In their instructions, manufacturers indicate the necessity of preventing embolic polymers from entering the venous compartment of circulation only due to the possibility of its thrombosis [31,32]. At the same time, it is seen that a rational approach to selecting embolic polymers, based on the diameter of the vessels of the racimous compartment, is also associated with hydrodynamic factors—excessive pressure can lead to rupture of the AVM. In the absence of large vessels in the AVM nidus, the values of the pressure maxima for both polymers almost completely coincide, and the shear stresses differ only when a certain threshold value of the diameter of small vessels (1–1.2 mm) is exceeded. In addition, it can be seen from the experimental data (Figure 13, Table 2), that the velocity in the racimous part of the AVM decreases significantly and the maximum flow velocity increases for both polymers due to fistula presents.

### 3.2. Governing Equations for Viscosity

It can be seen from Figure 14 that at both body and laboratory temperatures ($37{\;}^{{}^{\circ}}$C and $20{\;}^{{}^{\circ}}$C, respectively), the embolic polymers tend to behave like a pseudoplastic liquid with a non-zero flow limit. One of the simplest models of power–law dependence of viscosity on shear rate is the Ostwald–de Waele model. Its generalization of the spatial case of flow gives the following expression for the dependence of viscosity on shear rate [33]:(4)$$\mu_{eff}=kU^{n-1},$$
where *k*—fluid consistency index, *U*—shear rate, and *n*—degree of non-Newtonian behavior of a material.

However, the obvious disadvantage of such models is the behavior of viscosity at very large (*U* >> 1) or, conversely, very small (*U* << 1) shear rates. Cisco’s rheological model overcomes this shortcoming [34]: (5)$$\mu_{eff}=\mu_{0}+kU^{n-1}$$
where as $\mu_{0}$, the value of viscosity to which the experimental data tend to stay at high shear rates must be chosen.

From these types of experimental graphs (see Figure 14) and an understanding of the different shear rates for the flow in the catheter and in the rational part of the malformation, it is seen that the interesting shear rate ranges are 10–100 and 0.1–10 s${}^{-1}$. For the first interval, an almost constant value of viscosity is observed. For the second range of shear rates, the viscosity changes, as in a pseudoplastic fluid, increasing with decreasing shear rates.

This law is well approximated by the CM, and the approximation was made in the Wolfram Mathematica package (license of LIH SB RAS). The values of the empirical constants are given in Table 3.

### 3.3. Understanding of Viscosity Activation Process

The selected model approximates the experimental data quite well (see Figure 14) for the selected interval; however, when it is considered over a larger interval, its approximating ability drops significantly, so it is not recommended to use this model in the shear rate range >10 s${}^{-1}$. Indeed, the mechanism of viscosity for this non-Newtonian fluid at such values of shear rate is completely different, and the viscosity can be conditionally considered constant, although it depends significantly on temperature. Consequently, two interesting remarks can be made through analyzing the obtained experimental data.

First of all, a zone of nonmonotonic dependence of viscosity on shear rate at a value of the latter of about 1 s${}^{-1}$ can be noticed (see Figure 15). Moreover, it is seen that this nonmonotonicity is most noticeable precisely at physiological temperatures. This zone of nonmonotonicity can be given the following description. The fact is that the studied preparations are used to embolize a network of small vessels; however, between reaching this small network and leaving the microcatheter, the polymers, within a very short time, are in the vessel of a larger one or, in the case of inserting the tip of the microcatheter into the AVM nidus (which often has aneurysmal dilatations), of a significantly larger diameter, which causes the shear rate of the polymer to move through the region of nonmonotonicity and, as a consequence, a significant decrease in viscosity (about 30%). This, together with the understanding of the presence of vortex formation in the AVM nidus, gives confidence in a more uniform distribution of the polymer from the AVM nidus into the network of small vessels.

Secondly, it gives an understanding that insufficiently heated polymer should not be delivered to the AVM nidus, since its high viscosity can prevent uniform distribution among the network of small vessels exiting the AVM nidus. From Figure 15, it is seen that the Squid-12^®^ polymer is especially sensitive to this. In cases where the embolic polymer is used in neurosurgical operations, the length of the catheter from the site of catheter insertion (usually the femoral artery) to the site of embolic polymer injection is at least 1 m and, according to our numerical simulation (see Figure 16), the embolic polymer has time to warm up in the microcatheter before entering the blood. However, with other approaches and/or operations with infants, where the distance from the site of catheter insertion to the site of injection of the embolic polymer can be measured by only centimeters, one should take into account the fact that the embolic polymer may not completely warm up when it enters the bloodstream, which may affect the uniformity of filling small AVM vessels.

Thirdly, understanding the process of changing the viscosity of the embolic polymer during an operation (see Figure 17) allows us to conclude that the embolization procedure needs to be improved. The fact is that according to the scheme in Figure 17, there is always a bolus of cold embolic polymer, which has a sufficiently high viscosity, between the surgeon and the patient. In our opinion, if you warm up as much of the catheter as possible between the patient and the syringe, then the control of the embolization process should become more accurate, and many different complications should be avoided [35,36].

## 4. Discussion

From the point of view of fundamental hydrodynamics of complex media, this is the problem in describing the motion of a non-Newtonian fluid. The flow of a viscous fluid in a channel can occur in different modes, depending on the geometric dimensions of the channel and the rheological properties of the fluid. The task of embolization is multiparametric, and the optimal mode depends on several parameters: the cross section of the channel, the rate of its introduction and the rheology of the embolization polymer. The solution of this problem is intended to present a protocol to implement a process of filling with a polymer a geometrically complex, branched network of degenerate vessels of an arteriovenous malformation or tumor, subject to certain conditions that are dictated by the physiology of the blood circulation and the local anatomy of the vessels. In the task of embolizing an anomaly such as an arteriovenous malformation, the characteristic diameter of small vessels in the AVM is about 0.2–1 mm, and the Reynolds number in such channels for blood flow is small, which ensures the laminar nature of the blood flow [33]:$$Re=\frac{\rho uL}{\mu },$$
where ρ—density, *u*—flow rate, *L*—hydraulic diameter, and μ—dynamic viscosity.

The Reynolds number for embolic flow is of the same order at the AVM nidus and will be of a completely different order for catheter flow. Indeed, to proceed from the size of the catheter 6 Fr and the volume of embolization of 0.6 mL/min, then the speed of material in the catheter will be about 3 mm/s. This means that the magnitude of the shear rate, according to Figure 9, is about 10 s${}^{-1}$. Taking into account the results of numerical calculations (for example, for an AVM with a vessel diameter of the racimous part of 0.5 mm), the following values of the Reynolds number for the embolic polymers in the AVM and in the catheter will be
$$Re_{{squid}_{20}}=0.44,Re_{{squid}_{37}}=0.58,Re_{avm}=4.8,$$
where $Re_{{squid}_{20}}$ is the Reynolds number of the SQUID-12^®^ embolisate in the cold part of the catheter ($T=20{\;}^{{}^{\circ}}$C), $Re_{{squid}_{37}}$ is the Reynolds number in the heated part of the catheter ($T=37{\;}^{{}^{\circ}}$C), and $Re_{avm}$ is the Reynolds number for SQUID-12^®^ in the AVM nidus.

Thus, the task of determining the rheology of a polymer is key for its selection and the development of an optimal embolization scenario. This problem can be solved only with an integrated approach, using both experimental approaches to study the rheological properties of polymers, and mathematical and computer modeling to describe the embolization process. Qualitative assessments of the quantities of interest, the shape of the AVM vessels as round, and using the Darcy–Weisbach (DW) and Poiseuille formulas, can be proposed [37,38]:(6)$$h_{mp}=\lambda \frac{l}{d}\frac{c^{2}}{2g},$$
(7)$$\delta p=\frac{32\mu lc}{d^{2}},$$
where $h_{mp}$—head losses, δp—pressure drop, λ—Darcy friction factor, *l*—tube length, *c*—flow rate, *d*—tube diameter, *g*—local acceleration, and μ—dynamic viscosity.

The first is to calculate the pressure loss in the pipe during the flow of fluid. The second is to evaluate the pressure drop at the inlet and outlet, which is proportional to the viscosity of the fluid and inversely proportional to the square of the pipe diameter. The embolization process in the first approximation can be considered as quasi-stationary, since the polymer supply rate is rather low (see above). Therefore, at a constant pressure drop, which is provided by the surgeon by adjusting the polymer supply through the conductor, and the constancy of other system parameters (tube lengths and viscosity) in the DW formula, AVM vessels with a relatively large cross section will be filled first of all.

This will lead to a decrease in the effective area of the AVM vessels, and hence to an increase in the resistance of this node. To fill smaller AVM vessels, one must either increase the pressure drop or decrease the viscosity of the polymer. Its viscosity, in turn, depends on both the diameter of the vessel and the feed rate, in addition to, as it has been seen in the course of rheometric tests, temperature. It is different for the conditions of the operating room (the outer part of the catheter through which the embolic polymer is injected) and the circulatory zone, and also changes under the influence of heating of the catheter inside the body.

Given the complexity of the embolization procedure and preparation for it [31,32], the operating surgeon faces a challenge regarding the amount and method of administration of the material. The documentation of both studied preparations describes only borderline cases in which the administration of the embolization polymer should be stopped or, conversely, preparation begun for the start of embolization. Manufacturers do not give direct advice or protocols that would unambiguously regulate the embolization procedure of the company precisely because, as discussed above, the rheology of the polymer, the geometry of the flow area and the type of polymer are closely related, which makes it impossible to recommend one or the other protocol at the instruction level. It should be noted that attempts to develop such protocols as well as optimal embolization scenarios have been made in recent years.

The problem under consideration has several fundamentally difficult moments. Firstly, modern hydrodynamics does not have a complete solution to the problem of the flow of even a viscous Newtonian fluid through a bifurcation, or a branching of channels. For a stationary flow, there is only one law of conservation of mass, which does not require knowledge of the properties of the fluid and channel walls. In this case, the only mechanism of energy dissipation is viscous dissipation and vortex formation, which immediately complicates the flow structure, generating secondary flows. The fundamental question about the energy (structure) of such a flow, about the energy loss of the flow during the passage of the tee, is the key to controlling the fluid flow.

Orlowski et al. calculated the flow and observed the results of a numerical calculation of embolization of a two-dimensional cavity [19]; however, the authors did not draw any conclusions about the optimality of the injection process, since the optimal parameters were not introduced and the calculations were not compared with any clinical data. The flow area is a model and, again, does not map to a clinically significant area. The authors of this work used the power law of the dependence of viscosity on the shear modulus. In general, this correctly reflects the nature of rheological relationships. However, this approach is too simplistic, since, as seen from the test results, the coefficients in the rheological models describing them have different values for different polymers.

The first attempts at modelling the embolization process, in which pathological vessels overlap and further flow into a healthy vascular bed occurs, were based on Darcy’s law and the Maag formula [39]. Branched network-type AVM models have been described by Golovin et al. [40]. Furher, the concept of considering AVM nidus embolization as a model of two-component filtration, in which the displaced component was blood and the displacing component was embolization material, which is a Newtonian fluid, was formulated and implemented by Cherevko et al. [41]. Such a simplification is essential, but it allowed the authors to solve the problem of optimal control of both single-stage (total) embolization and multi-stage embolization, when subtotal embolization is performed at all stages except the last one [42]. The use of rheological relations (Table 3) with a known set of constants for the two tested polymers will make it possible to more accurately formulate the law of optimal embolization.

In some papers, the clinical aspects of the use of both embolic polymers were evaluated in detail, and their brief chemical characteristics were given [43,44]. A review of the chemical properties of both the aforementioned embolic polymers and promising embolization hydrogels was carried out [26]. The study of their rheological properties is also an important task for the development of correct recommended protocols.

It should be noted that there are a number of aspects in the study that require further study. Further studies would make it possible to more accurately determine the laws of dependence of the rheology of embolization polymers on temperature. So far, this study has been carried out for a small temperature range. One of the explanations for this limitation is the difficulty in supplying these polymers for laboratory research, since the circulation of such polymers in the Russian Federation is allowed only through medical institutions. In medical institutions, these polymers are strictly accounted for and consumed exclusively for endovascular operations, so such substances can only get into the research laboratory as a “waste of operational activities”. This leads to the fact that the collection of the amount of polymers necessary for all rheological tests took quite a long time. It should be noted that rheological tests of the Phil polymer have also been carried out, but at the moment enough “waste” of this polymer to carry out the entire line of tests has not been collected. Another limitation in our work is the assumption of a model AVM configuration for numerical calculation.

However, a number of authors [2,41] have worked with this kind of model configuration, which reflects the main patterns of fluid flow in a complex network of vessels. Many authors have tried to get a visualization of the flow of embolic polymer inside the AVM nidus [9,19], but so far this activity is associated with significant difficulties. In our opinion, the ultimate goal of modeling is not to reveal the picture of the flow of embolic polymers, but to obtain more general results; such an approach would help to build a model for the optimal selection of an embolization polymer for a particular clinical case and the optimal method for its administration.

## 5. Conclusions

In the framework of this study, rheological measurement tests of two non-adhesive polymers (Onyx-18^®^ and Squid-12^®^) used in medicine for AVM embolization were performed. The analysis of the results of these tests showed a significant dependence of the viscosity not only on the shear modulus but also on the temperature. In the course of numerical simulation of model embolization of a branched network of vessels, different behavior of these embolic polymers for fistula and racimous network models was demonstrated, which confirms the need to use embolic polymers of different viscosities for embolization of these AVM parts. Of particular interest is the physical interpretation of the law of dependence of the viscosity of embolic polymers and the mechanism of thermal and shear activation in the AVM nidus. The found non-monotonicity of the dependence of viscosity on shear rate gives a new idea of the non-randomness of this effect and, in general, the viscosity activation algorithm with a change in the viscosity of the embolic polymer opens up a qualitative understanding of the effects that arise during embolization. In addition, this understanding suggests ways to solve problems such as difficult pumping of embolic polymer through a small microcatheter, as well as performing operations on infants by controlling the temperature in the outer (outside the body) part of the catheter. The presented coefficients of the Cisco model will allow numerical hemodynamic specialists to perform calculations in the case of a simultaneous change in shear rate and temperature. 

## Figures and Tables

**Figure 1 polymers-15-01060-f001:**
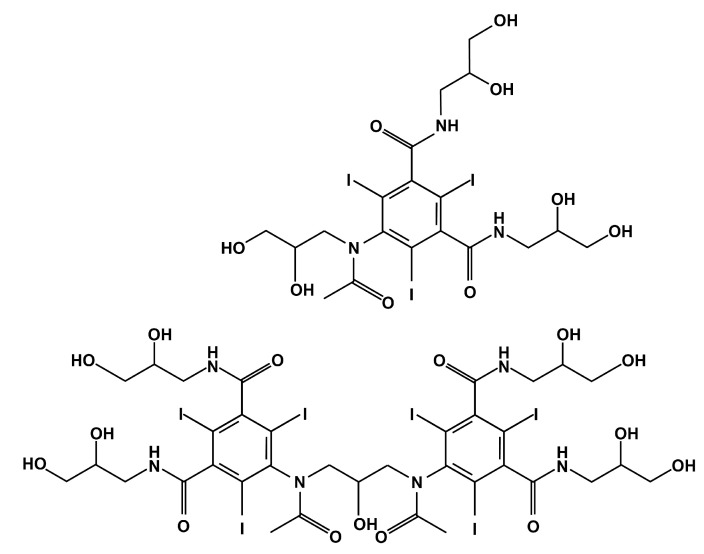
Structures of iodine-containing radiopaque substances.

**Figure 2 polymers-15-01060-f002:**
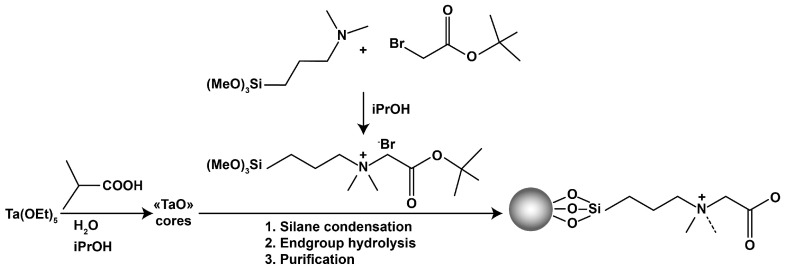
One of the mechanisms for creating tantalum powder for preparing a suspension consisting of an embolizing polymer and powder [17,22].

**Figure 3 polymers-15-01060-f003:**
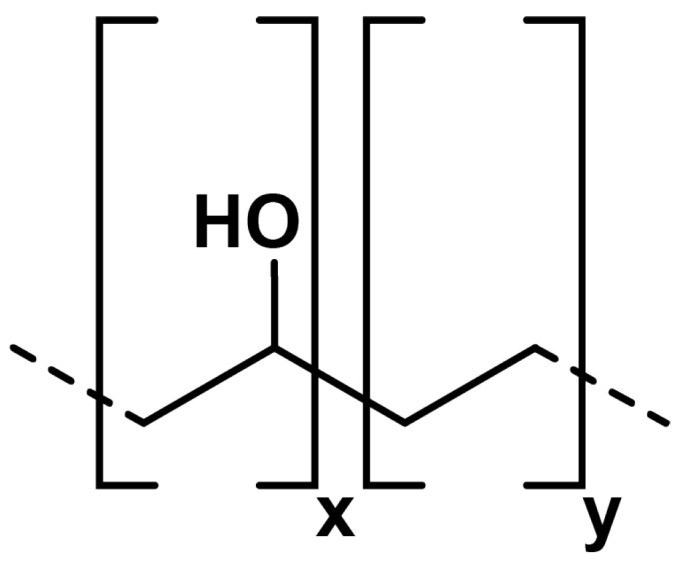
Schematic chemical formula of an ONYX-type embolic polymer [26].

**Figure 4 polymers-15-01060-f004:**
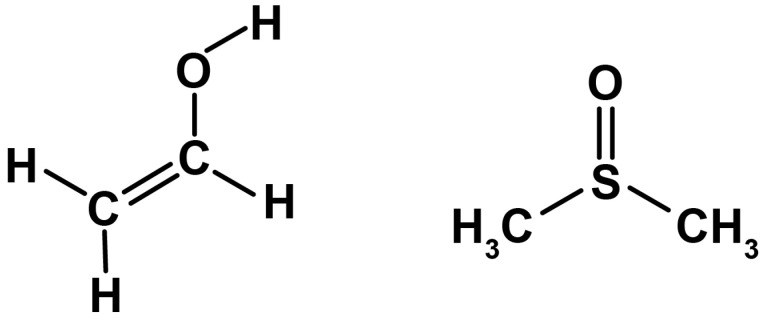
The structure of vinyl alcohol (**left**); the structure of DMSO (**right**).

**Figure 5 polymers-15-01060-f005:**
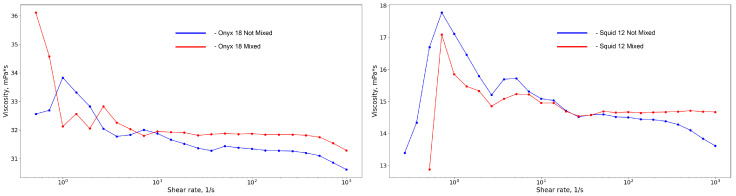
Experimental data of the viscosity dependence on shear rate of Onyx^®^ (**left**) and Squid^®^ (**right**) mixed and not mixed with tanthalum powder.

**Figure 6 polymers-15-01060-f006:**
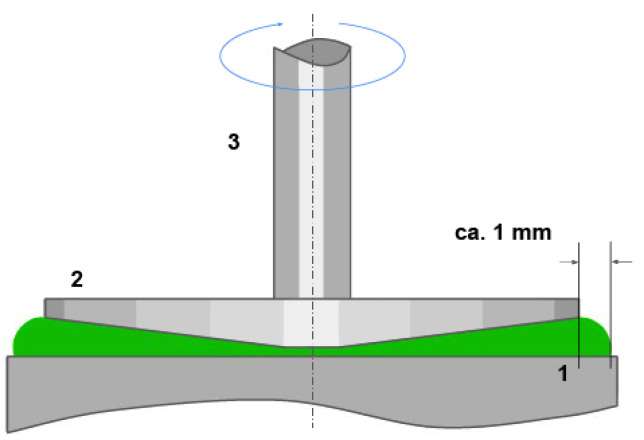
Illustrative scheme for testing using a cone-plane system [27]: 1: base of the measuring system; 2: device of the measuring system (in this case, the cone-plane); 3: shaft leading to the gas bearing of the measuring system.

**Figure 7 polymers-15-01060-f007:**
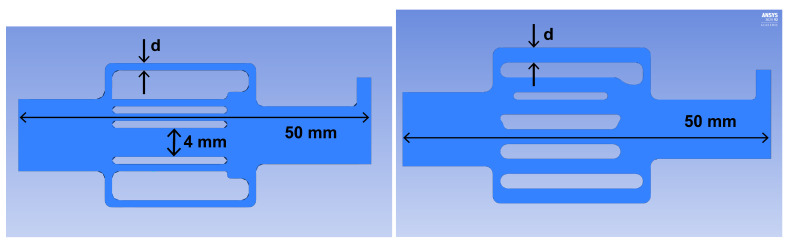
Geometry used for numerical calculations: AVM without fistula (**left**) with varying vessel diameter d∈(0.5,2) mm, AVM with fistula (**right**) 4 mm in diameter and varying vessel diameter d∈(0.5,2) mm.

**Figure 8 polymers-15-01060-f008:**
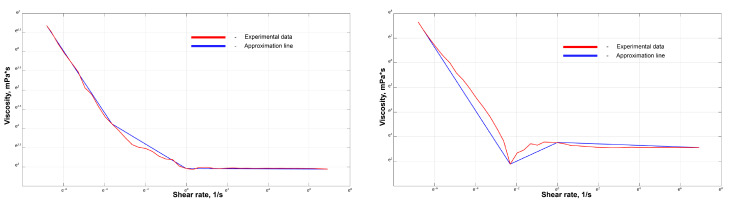
Experimental data of Onyx^®^ (**left**) and Squid^®^ (**right**) viscosity dependence on shear rate and the piecewise linear approximation that was used for numerical simulations (logarithmic scale).

**Figure 9 polymers-15-01060-f009:**
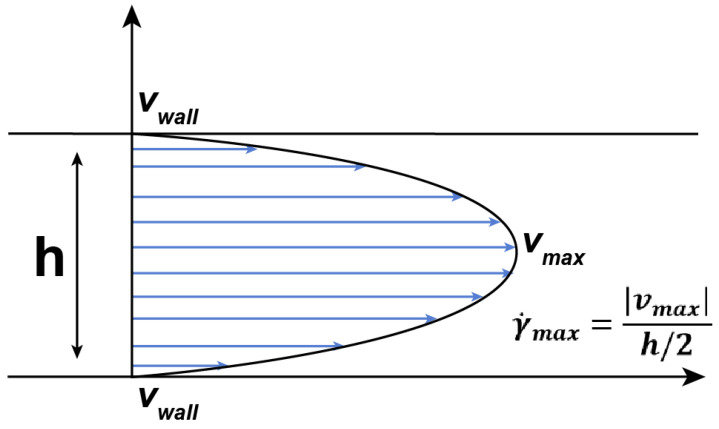
Poiseuille profile of the fluid flow and approximate shear rate.

**Figure 10 polymers-15-01060-f010:**
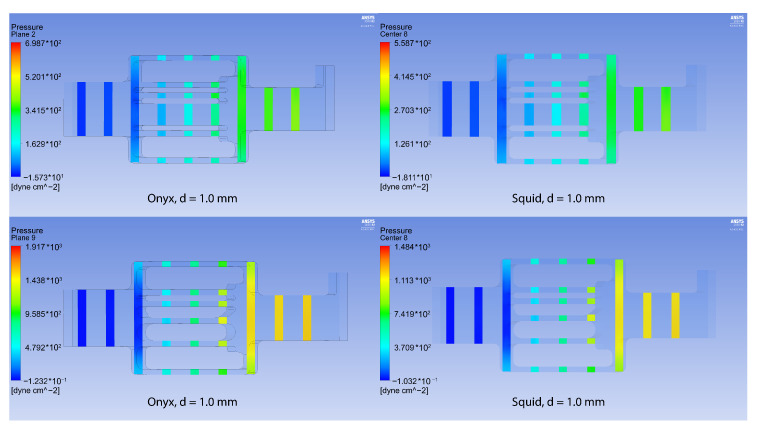
Pressure for the configurations with Onyx^®^ (**left column**) and Squid^®^ (**right column**) with fistula (**above**) and without it (**below**).

**Figure 11 polymers-15-01060-f011:**
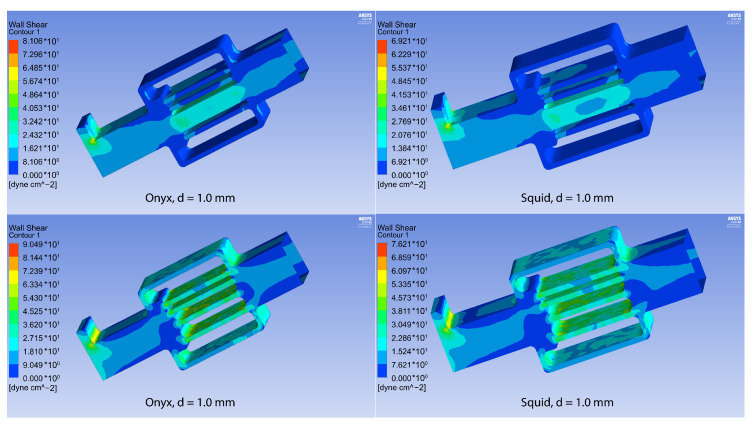
WSS for the configurations with Onyx^®^ (**left column**) and Squid^®^ (**right column**) with fistula (**above**) and without it (**below**).

**Figure 12 polymers-15-01060-f012:**
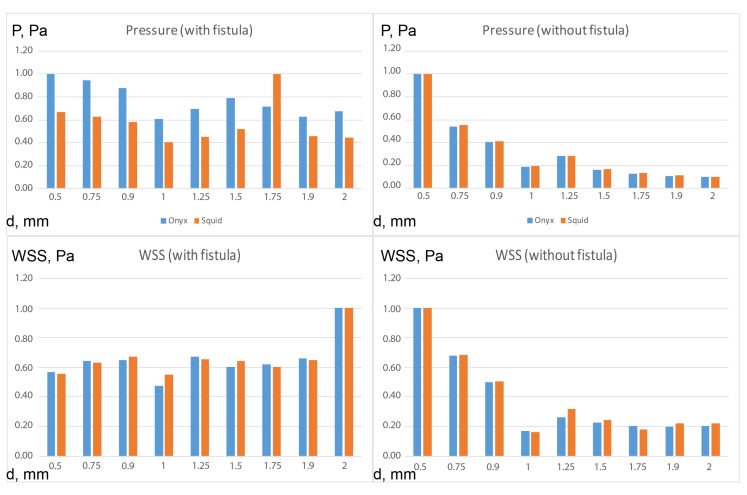
Pressure and WSS results of numerical calculations in the configuration with and without fistula for all considered vessel radii.

**Figure 13 polymers-15-01060-f013:**
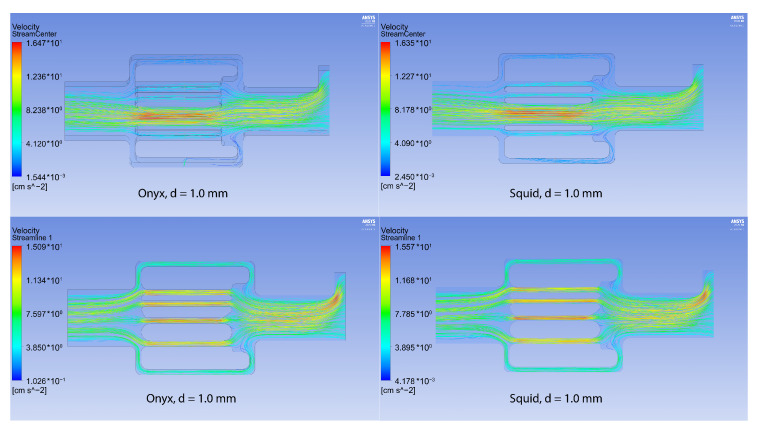
Velocity streamlines for the configurations with Onyx^®^ (**left column**) and Squid^®^ (**right column**) with fistula (**above**) and without it (**below**).

**Figure 14 polymers-15-01060-f014:**
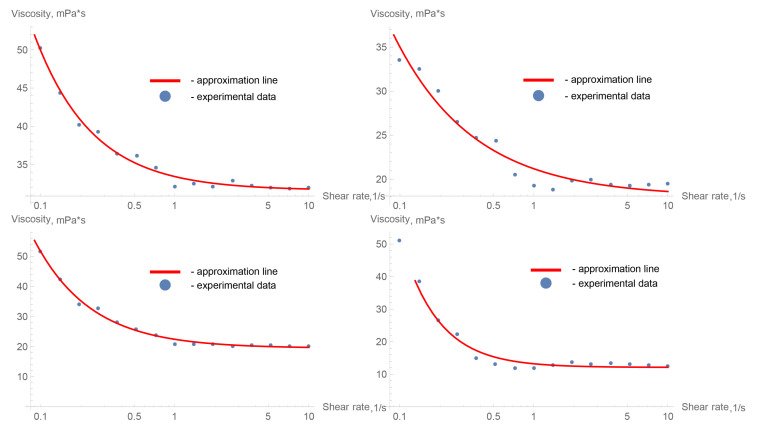
Approximation of experimental data of viscosity dependence on shear rate by Cisco’s model (CM) for the Onyx-18^®^ embolic polymer (**top**) and the Squid-12^®^ embolic polymer (**bottom**); for laboratory temperature $T=20{\;}^{{}^{\circ}}$C (**left**) and for physiological conditions $T=37{\;}^{{}^{\circ}}$C (**right**).

**Figure 15 polymers-15-01060-f015:**
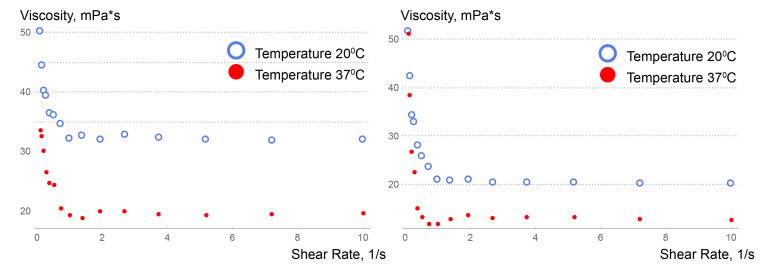
Comparison of experimental data on viscosity versus shear rate for Onyx-18^®^ (**left**) and Squid-12^®^ (**right**) embolic polymers.

**Figure 16 polymers-15-01060-f016:**
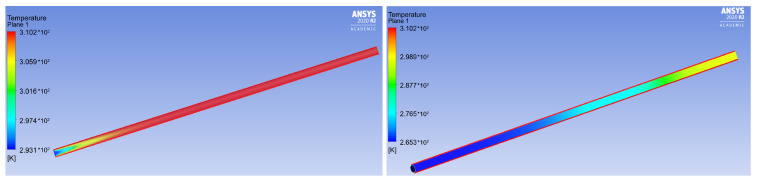
Temperature distribution in the catheter with a non-Newtonian (**left**) and Newtonian (**right**) model of Squid-12^®^ viscosity.

**Figure 17 polymers-15-01060-f017:**
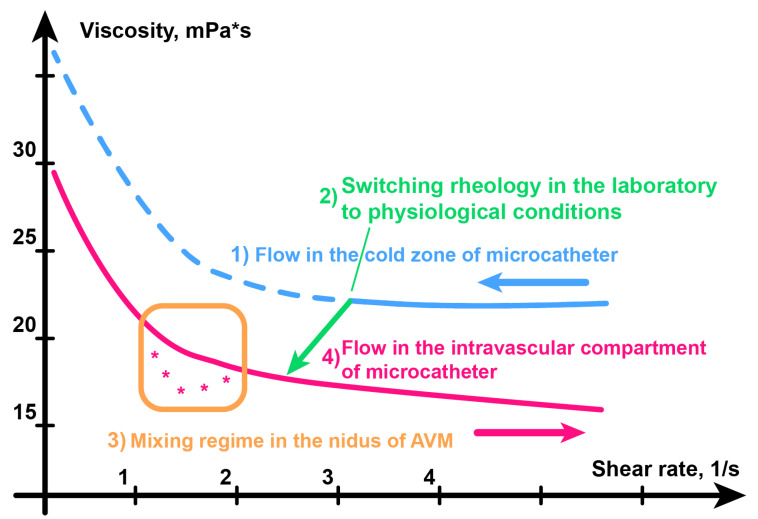
Viscosity change mechanism during embolization.

**Table 1 polymers-15-01060-t001:** Pressure and WSS results of numerical calculations in the configuration with and without fistula.

Diameter of the Vessel Racimous Part, mm	Fistula	Maximum Pressure in Racimous Part, Pa		Maximum WSS in Racimous Part, dyne/cm${}^{2}$	
		Onyx-18 ^®^	Squid-12 ^®^	Onyx-18 ^®^	Squid-12 ^®^
0.5	−	7665.17	5588.45	441.79	329.42
0.75	−	4148.71	3083.56	298.36	225.19
0.9	−	3076.53	2290.17	220.04	166.06
1	−	1432.47	1083.3	74.77	54.32
1.25	−	2146.94	1590.51	115.62	105.26
1.5	−	1246.53	938.47	100.59	80.44
1.75	−	970.86	735.596	89.33	59.29
1.9	−	813.85	617.49	87.04	72.96
2	−	743.36	565.05	89.49	73.23
0.5	+	560.99	432.79	29.61	24.05
0.75	+	529.44	406.56	33.53	27.20
0.9	+	492.31	376.38	33.80	28.91
1	+	340.15	261.82	24.83	23.65
1.25	+	389.53	294.17	34.85	28.13
1.5	+	443.49	338.72	31.43	27.85
1.75	+	400.98	649.34	32.13	26.01
1.9	+	351.55	295.15	34.19	28.08
2	+	378.34	288.94	52.07	43.20

**Table 2 polymers-15-01060-t002:** Velocity and shear rate results of numerical calculations in the configuration with and without fistula.

	Onyx-18^®^			Squid-12^®^		
**d = 1 mm**	**Max Velocity**	**Min Velocity**	**Shear Rate**	**Max Velocity**	**Min Velocity**	**Shear Rate**
Fistula	0.15	1647.78	29.62	0.24	1635.40	30.83
Without fistula	0.25	1548.08	45.23	0.41	1556.58	46.60

**Table 3 polymers-15-01060-t003:** Values of empirical constants of the CM for the studied embolic polymers at laboratory ($20{\;}^{{}^{\circ}}$C) and physiological ($37{\;}^{{}^{\circ}}$C) temperatures.

Embolic Polymer	$T=20{\;}^{{}^{\circ}}$C			$T=37{\;}^{{}^{\circ}}$C		
	$\mu_{0}$	k	n	$\mu_{0}$	k	n
Onyx-18^®^	31.6187	1.80968	−0.004368	18.0138	3.17306	0.271126
Squid-12^®^	19.4711	2.9736	−0.03329	12.1248	1.10875	−0.5586

## Data Availability

Experimental data for considered shear-rate intervals is attached. Full experimental data and results of numerical simulations are available upon request.

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
