# Peer review of "Rheological Properties of Non-Adhesive Embolizing Compounds—The Key to Fine-Tuning Embolization Process-Modeling in Endovascular Surgery"

_polymers, 2023, doi:10.3390/polym15041060_

Round 1

Reviewer 1 Report

In this study, two polymers used for embolization of arteriovenous malformations (AVMs, Onyx-18 ® and Squid- 12®) are studied through rheological tests. The law of the dependence of viscosity on temperature in the range from 20C to 37C was established. Differences in the hydrodynamic characteristics of both polymers were studied using numerical simulation in model configurations of AVM. The study was well-designed and includes findings with interests. The following issues should be considered.

1.       Please avoid using unconvincing words, like “for the first time”, “extremely”...

2.       The study mainly focuses on the “non-adhesive embolizing compounds”, why? Will these embolization agents flow into the neighboring tissue and cause the necrosis of normal tissue? A comparison and discussion is necessary between adhesive and non-adhesive embolizing compounds. A recent study on adhesive embolizing agent “Development of PVA-based microsphere as a potential embolization agent” is recommended to cite.

3.       Some common information about the used embolizing compounds (Onyx-18 ® and Squid- 12®) should be given, like the size…

4.       This paper contains a lot of equations, if not common sense (like Reynolds number), please give the citation to specify their origin.

5.       The layout of citation is strange (line 304 - line 307: in [23], in [24], in [25]…), and should follow the journal instructions.

Reviewer 2 Report

Dear Authors, I have some concerns to arise:

Abstract: the structure of this section could be improved. It is difficult to understand the scenario of this study, the key points and so on. Please ensure a structural abstract.

Introduction: Line 18-31, Delete this part, it is a generic introduction useful for any kind of paper.

At the end of the introduction you could better clarify the end points of this study protocol.

Discussion: my compliments for your honest declaration about limitations of the study, probably this part is too long 
